# Neural Manifold Ordinary Differential Equations

**Aaron Lou\*, Derek Lim\*, Isay Katsman\*, Leo Huang\*, Qingxuan Jiang**
Cornell University
{al968, dl772, isk22, ah839, qj46}@cornell.edu

**Ser-Nam Lim**
Facebook AI
sernam@gmail.com

**Christopher De Sa**
Cornell University
cdesa@cs.cornell.edu

## Abstract

To better conform to data geometry, recent deep generative modelling techniques adapt Euclidean constructions to non-Euclidean spaces. In this paper, we study normalizing flows on manifolds. Previous work has developed flow models for specific cases; however, these advancements hand craft layers on a manifold-by-manifold basis, restricting generality and inducing cumbersome design constraints. We overcome these issues by introducing Neural Manifold Ordinary Differential Equations, a manifold generalization of Neural ODEs, which enables the construction of Manifold Continuous Normalizing Flows (MCNFs). MCNFs require only local geometry (therefore generalizing to arbitrary manifolds) and compute probabilities with continuous change of variables (allowing for a simple and expressive flow construction). We find that leveraging continuous manifold dynamics produces a marked improvement for both density estimation and downstream tasks.

## 1 Introduction

Deep generative models are a powerful class of neural networks which fit a probability distribution to produce new, unique samples. While latent variable models such as Generative Adversarial Networks (GANs) [18] and Variational Autoencoders (VAEs) [27] are capable of producing reasonable samples, computing the exact modeled data posterior is fundamentally intractable. By comparison, normalizing flows [38] are capable of learning rich and tractable posteriors by transforming a simple probability distribution through a sequence of invertible mappings. Formally, in a normalizing flow, a complex distribution $p(x)$ is transformed to a simple distribution $\pi(z)$ via a diffeomorphism $f$ (i.e. a differentiable bijective map with a differentiable inverse) with probability values given by the change of variables:

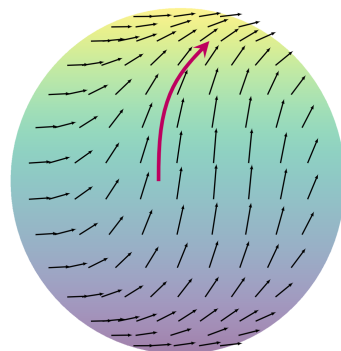

Figure 1: A manifold ODE solution for a given vector field on the sphere.

$$\log p(x) = \log \pi(z) - \log \det \left| \frac{\partial f^{-1}}{\partial z} \right|, \qquad z = f(x).$$

To compute this update efficiently, $f$ must be constrained to allow for fast evaluation of the determinant, which in the absence of additional constraints takes

$\mathcal{O}(D^3)$ time (where $D$ is the dimension of $z$). Furthermore, to efficiently generate samples, $f$ must have a computationally cheap inverse. Existing literature increases the expressiveness of such models under these computational constraints and oftentimes parameterizes $f$ with deep neural networks [17, 8, 26, 11]. An important recent advancement, dubbed the Continuous Normalizing Flow (CNF), constructs $f$ using a Neural Ordinary Differential Equation (ODE) with dynamics $g$ and invokes a continuous change of variables which requires only the trace of the Jacobian of $g$ [4, 19].

Since $f$ is a diffeomorphism, the topologies of the distributions $p$ and $\pi$ must be equivalent. Furthermore, this topology must conform with the underlying latent space, which previous work mostly assumes to be Euclidean. However, topologically nontrivial data often arise in real world examples such as in quantum field theory [45], motion estimation [14], and protein structure prediction [22].

In order to go beyond topologically trivial Euclidean space, one can model the latent space with a *smooth manifold*. An $n$-dimensional manifold $\mathcal{M}$ can be thought of as an $n$-dimensional analogue of a surface. Concretely, this is formalized with *charts*, which are smooth bijections $\varphi_x : U_x \to V_x$, where $U_x \subseteq \mathbb{R}^n, x \in V_x \subseteq \mathcal{M}$, that also satisfy a smoothness condition when passing between charts. For charts $\varphi_{x_1}, \varphi_{x_2}$ with corresponding $U_{x_1}, V_{x_1}, U_{x_2}, V_{x_2}$, the composed map $\varphi_{x_2}^{-1} \circ \varphi_{x_1} : \varphi_{x_1}^{-1}(V_{x_1} \cap V_{x_2}) \to \varphi_{x_2}^{-1}(V_{x_1} \cap V_{x_2})$ is a diffeomorphism.

Preexisting manifold normalizing flow works (which we present a complete history of in Section 2) do not generalize to arbitrary manifolds. Furthermore, many examples present constructions extrinsic to the manifold. In this work, we solve these issues by introducing Manifold Continuous Normalizing Flows (MCNFs), a manifold analogue of Continuous Normalizing Flows. Concretely, we:

(i) introduce Neural Manifold ODEs as a generalization of Neural ODEs (seen in Figure 1). We leverage existing literature to provide methods for forward mode integration, and we derive a manifold analogue of the adjoint method [37] for backward mode gradient computation.

(ii) develop a dynamic chart method to realize Neural Manifold ODEs in practice. This approach integrates local dynamics in Euclidean space and passes between domains using smooth chart transitions. Because of this, we perform computations efficiently and can accurately compute gradients. Additionally, this allows us to access advanced ODE solvers (without manifold equivalents) and augment the Neural Manifold ODE with existing Neural ODE improvements [10, 19].

(iii) construct Manifold Continuous Normalizing Flows. These flows are constructed by integrating local dynamics to construct diffeomorphisms, meaning that they are theoretically complete over general manifolds. Empirically, we find that our method outperforms existing manifold normalizing flows on their specific domain.

## 2 Related Work

In this section we analyze all major preexisting manifold normalizing flows. Previous methods are, in general, hampered by a lack of generality and are burdensomely constructive.

**Normalizing Flows on Riemannian Manifolds [16].** The first manifold normalizing flow work constructs examples on Riemannian manifolds by first projecting onto Euclidean space, applying a predefined Euclidean normalizing flow, and projecting back. Although simple, this construction is theoretically flawed since the initial manifold projection requires the manifold to be diffeomorphic to Euclidean space. This is not always the case, since, for example, the existence of antipodal points on a sphere necessarily implies that the sphere is not diffeomorphic to Euclidean space. As a result, the construction only works on a relatively small and topologically trivial subset of manifolds.

Our work overcomes this problem by integrating local dynamics to construct a global diffeomorphism. By doing so, we do not have to relate our entire manifold with some Euclidean space, but rather only well-behaved local neighborhoods. We test against [16] on hyperbolic space, and our results produce a significant improvement.

**Latent Variable Modeling with Hyperbolic Normalizing Flows [2].** In a recent manifold normalizing flow paper, the authors propose two normalizing flows on hyperbolic space—a specific

---

All of our manifolds are assumed to be smooth, so we refer to them simply as manifolds.

This example is generalized by the notion of conjugate points. Most manifolds have conjugate points and those without are topologically equivalent to Euclidean space.

Riemannian manifold. These models, which they name the Tangent Coupling (TC) and Wrapped Hyperboloid Coupling (WHC), are not affected by the aforementioned problem since hyperbolic space is diffeomorphic to Euclidean space. However, various shortcomings exist. First, in our experiments we find that the methods do not seem to conclusively outperform [16]. Second, these methods do not generalize to topologically nontrivial manifolds. This means that these flows produce no additional topological complexity and thus the main benefit of manifold normalizing flows is not realized. Third, the WHC construction is not intrinsic to hyperbolic space since it relies on the hyperboloid equation.

Our method, by contrast, is derived from vector fields, which are natural manifold constructs. This not only allows for generality, but also means that our construction respects the manifold geometry. We compare against [2] on hyperbolic space and find that our results produce a substantial improvement.

**Normalizing Flows on Tori and Spheres [39].** In another paper, the authors introduce a variety of normalizing flows for tori and spheres. These manifolds are not diffeomorphic to Euclidean space, hence the authors construct explicit global diffeomorphisms. However, their constructions do not generalize and must be intricately designed with manifold topology in mind. In addition, the primary recursive $\mathbb{S}^n$ flow makes use of non-global diffeomorphisms to the cylinder, so densities are not defined everywhere. The secondary exponential map-based $\mathbb{S}^n$ flow is globally defined but is not computationally tractable for higher dimensions.

In comparison, our work is general, requires only local diffeomorphisms, produces globally defined densities, and is tractable for higher dimensions. We test against [39] on the sphere and attain markedly better results.

**Other Related Work.** In [12], the authors define a probability distribution on Lie Groups, a special type of manifold, and as a by-product construct a normalizing flow. The constructed flow is very similar to that found in [16], but the authors include a $\tanh$ nonlinearity at the end of the Euclidean flow to constrain the input space and guarantee that the map back to the manifold is injective. We do not compare directly against this work since Lie Groups are not general (the 2-dimensional sphere is not a Lie Group [41]) while Riemannian manifolds are.

There are some related works such as [45, 43, 3] that are orthogonal to our considerations as they either (i) develop applications as opposed to theory or (ii) utilize normalizing flows as a tool to study Riemannian metrics.

Concurrent work [35, 13] also investigates the extension of neural ODEs to smoooth manifolds. Our work introduces the dynamic chart method to better evaluate and differentiate through the manifold ODEs; [35] does not develop an adjoint method and [13] differentiates in the ambient space similar to our Section 4.2. Both of these works calculate the change in log probability with a Riemannian change of variables, while we calculate this change through the use of charts and Euclidean change of variables.

## 3 Background

In this section, we present background knowledge to establish naming conventions and intuitively illustrate the constructions used for our work. For a more detailed overview, we recommend consulting a text such as [31, 32, 9].

### 3.1 Differential Geometry

**Tangent space.** For an $n$-dimensional manifold $\mathcal{M}$, the *tangent space $T_x\mathcal{M}$* at a point $x \in \mathcal{M}$ is a higher-dimensional analogue of a tangent plane at a point on a surface. It is an $n$-dimensional real vector space for all points $x \in \mathcal{M}$.

For our purposes, tangent spaces will play two roles. First, they provide a notion of derivation which is crucial in defining manifold ODEs. Second, they will oftentimes be used in place of $\mathbb{R}^n$ for our charts (as we map $U_x$ to some open subset of $T_x\mathcal{M}$ through a change of basis from $\mathbb{R}^n \to T_x\mathcal{M}$).

**Pushforward/Differential.** A derivative (or a *pushforward*) of a function $f : \mathcal{M} \to \mathcal{N}$ between two manifolds is denoted by $D_x f : T_x\mathcal{M} \to T_x\mathcal{N}$. This is a generalization of the classical Euclidean Jacobian (as $\mathbb{R}^n$ is a manifold), and provides a way to relate tangent spaces at different points.

As one might expect, the pushforward is central in the definition of manifold ODEs (analogous to the importance of the common derivative in Euclidean ODEs). We also use it in our dynamic chart method to map tangent vectors of the manifold to tangent vectors of Euclidean space.

## 3.2 Riemannian Geometry

While the above theory is general, to concretize some computational aspects (e.g. how to pick charts) and give background on related manifold normalizing flow work, we define relevant concepts from Riemannian geometry.

**Riemannian Manifold.** The fundamental object of study in Riemannian geometry is the *Riemannian manifold*. This is a manifold with an additional structure called the *Riemannian metric*, which is a smooth metric $\rho_x : T_x\mathcal{M} \times T_x\mathcal{M} \to \mathbb{R}$ (often denoted as $\langle \cdot, \cdot \rangle_\rho$). This Riemannian metric allows us to construct a distance on the manifold $d_\rho : \mathcal{M} \times \mathcal{M} \to \mathbb{R}$. Furthermore, any manifold can be given a Riemannian metric.

**Exponential Map.** The *exponential map* $\exp_x : T_x\mathcal{M} \to \mathcal{M}$ can be thought of as taking a vector $v \in T_x\mathcal{M}$ and following the general direction (on the manifold) such that the distance traveled is the length of the tangent vector. Specifically, the distance on the manifold matches the induced tangent space norm $d_\rho(x, \exp_x(v)) = \|v\|_\rho := \sqrt{\langle v, v \rangle_\rho}$. Note that $\exp_x(0) = x$.

The exponential map is crucial in our construction since it acts as a chart. Specifically, if we identify the chart domain with $T_x\mathcal{M}$ then $\exp_x$ is a diffeomorphism when restricted to some local set around $0$.

**Special Cases.** Some special cases of Riemannian manifolds include hyperbolic spaces $\mathbb{H}^n = \{x \in \mathbb{R}^{n+1} : -x_1^2 + \sum_{i=2}^{n+1} x_i^2 = -1, \ x_1 > 0\}$, spheres $\mathbb{S}^n = \{x \in \mathbb{R}^{n+1} : \sum_{i=1}^{n+1} x_i^2 = 1\}$, and tori $\mathbb{T}^n = (\mathbb{S}^1)^n$. Specific formulas for Riemannian computations are given in Appendix B. Hyperbolic space is diffeomorphic to Euclidean space, but spheres and tori are not.

## 3.3 Manifold Ordinary Differential Equations

**Manifold ODE.** Finally, we introduce the key objects of study: manifold ordinary differential equations. A manifold ODE is an equation which relates a curve $\mathbf{z} : [t_s, t_e] \to \mathcal{M}$ to a vector field $f$ and takes the form

$$\frac{d\mathbf{z}(t)}{dt} = f(\mathbf{z}(t), t) \in T_{\mathbf{z}(t)}\mathcal{M} \qquad \mathbf{z}(t_s) = z_s \tag{1}$$

$\mathbf{z}$ is a *solution* to the ODE if it satisfies Equation 1 with initial condition $z_s$. Similarly to the case of classical ODEs, local solutions are guaranteed to exist under sufficient conditions on $f$ [20].

# 4 Neural Manifold Ordinary Differential Equations

To leverage manifold ODEs in a deep learning framework similar to Neural ODEs [4], we parameterize the dynamics $f$ by a neural network with parameters $\theta$. We define both forward pass and backward pass gradient computation strategies in this framework. Unlike Neural ODEs, forward and backward computations do not perfectly mirror each other since forward mode computation requires explicit manifold methods, while the backward can be defined solely through an ODE in Euclidean space.

## 4.1 Forward Mode Integration

The first step in defining our Neural Manifold ODE block is the forward mode integration. We review and select appropriate solvers from the literature; for a more thorough introduction we recommend consulting a text such as [5, 20]. Broadly speaking, these solvers can be classified into two groups:

(i) *projection methods* that embed the manifold into Euclidean space $\mathbb{R}^d$, integrate with some base Euclidean solver, and project to the manifold after each step. Projection methods require additional manifold structure; in particular, $\mathcal{M}$ must be the level set of some smooth function $g : \mathbb{R}^d \to \mathbb{R}$.

(ii) *implicit methods* that solve the manifold ODE locally using charts. These methods only require manifold-implicit constructions.

Projection methods are conceptually simple, but ultimately suffer from generality issues. In particular, for manifolds such as the open ball or the upper half-space, there is no principled way to project off-manifold points back on. Furthermore, even in nominally well-defined cases such as the sphere, there may still exist points such as the origin for which the projection is not well-defined.

Implicit methods, by contrast, can be applied to any manifold and do not require a level set representation. Thus, this approach is more amenable to our generality concerns (especially since we wish to work with, for example, hyperbolic space). However, difficulty in defining charts restricts applicability. Due to this reason, implicit schemes often employ additional structure to define manifold variations of step-based solvers [1, 5]. For example, on a Riemannian manifold one can define a variant of an Euler Method solver with update step $z_{t+\epsilon} = \exp_{z_t}(\epsilon f(z_t, t))$ using the Riemannian exponential map. On a Lie group, there are more advanced Runge-Kutta solvers that use the Lie Exponential map and coordinate frames [5, 20].

## 4.2 Backward Mode Adjoint Gradient Computation

In order to fully incorporate manifold ODEs into the deep learning framework, we must also efficiently compute gradients. Similar to [4], we develop an adjoint method to analytically calculate the derivative of a manifold ODE instead of directly differentiating through the solver. Similar to manifold adjoint methods for partial differential equations, we use an ambient space [46].

**Theorem 4.1.** *Suppose we have some manifold ODE as given in Equation 1 and we define some loss function $L : \mathcal{M} \to \mathbb{R}$. Suppose that there is an embedding of $\mathcal{M}$ in some Euclidean space $\mathbb{R}^d$ and we identify $T_x\mathcal{M}$ with an $n$-dimensional subspace of $\mathbb{R}^d$. If we define the adjoint state to be $\mathbf{a}(t) = D_{\mathbf{z}(t)}L$, then the adjoint satisfies*

$$\frac{d\mathbf{a}(t)}{dt} = -\mathbf{a}(t)D_{\mathbf{z}(t)}f(\mathbf{z}(t), t) \tag{2}$$

**Remark.** *This theorem resembles the adjoint method in [4] precisely because of our ambient space condition. In particular, our curve $\mathbf{z} : [t_s, t_e] \to \mathcal{M}$ can be considered as a curve in the ambient space. Furthermore, we do not lose any generality since such an embedding always exists by the Whitney Embedding Theorem [44], if we simply set $d = 2n$.*

The full proof of Theorem 4.1 can be found in Appendix A.3. Through this adjoint state, gradients can be derived for other parameters in the equation such as $t_s, t_e$, initial condition $z_s$, and weights $\theta$.

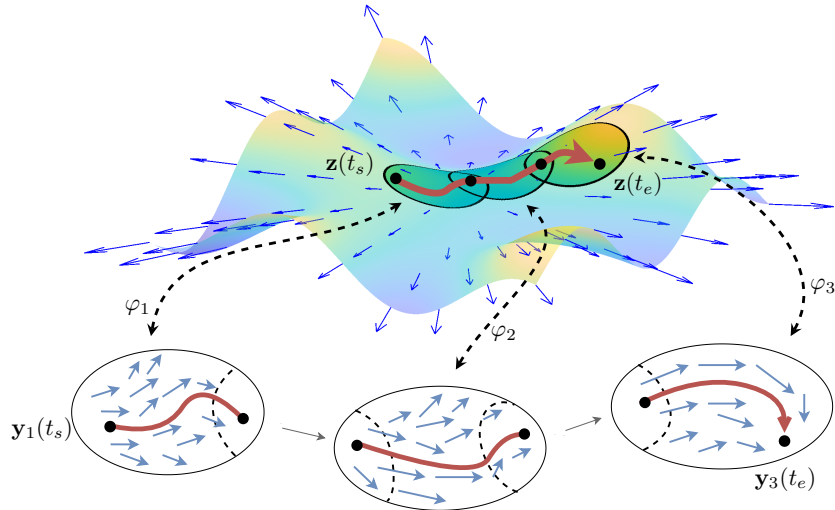

Figure 2: Solving a manifold ODE with our dynamic chart method. We use 3 charts.

## 5 Dynamic Chart Method

While our above theory is fully expressive and general, in this section we address certain computational issues and augment applicability with our *dynamic chart method*.

---
**Algorithm 1:** Dynamic Chart Forward Pass
---

Given $f$, local charts $\varphi_x$, starting condition $z_s$ and starting/ending times of $t_s, t_e$.
Initialize $z \leftarrow z_s, \tau \leftarrow t_s$
**while** $\tau < t_e$ **do**

    Construct an equivalent differential equation $\frac{d\mathbf{y}(t)}{dt} = D_{\varphi_z(\mathbf{y}(t))}\varphi_z^{-1} \circ f(\varphi_z(\mathbf{y}(t)), t)$ with
      initial condition $\mathbf{y}(\tau) = \varphi_z^{-1}(z)$
    Solve $\mathbf{y}$ locally using some numerical integration technique in Euclidean space. Specifically,
      solve in some interval $[\tau, \tau + \epsilon]$ for which $\mathbf{z}([\tau, \tau + \epsilon]) \subseteq \text{im}\varphi_z$.
    $z \leftarrow \varphi_z(\mathbf{y}(\tau + \epsilon)), \tau \leftarrow \tau + \epsilon$

**end**
---

The motivation for our dynamic chart method comes from [33], where the author introduces a *dynamic manifold trivialization* technique for Riemannian gradient descent. Here, instead of applying the traditional Riemannian gradient update $z_{t+1} = \exp_{z_t}(-\eta \nabla_{z_t} f)$ for $N$ time steps, the author repeatedly applies $n \ll N$ local updates. Each update consists of a local diffeomorphism to Euclidean space, $n$ equivalent Euclidean gradient updates, and a map back to the manifold. This allows us to lower the number of expensive exponential map calls and invoke existing Euclidean gradient optimizers such as Adam [25]. This is similar in spirit to [16], but crucially only relies on local diffeomorphisms rather than a global diffeomorphism.

We can adopt this strategy in our Neural Manifold ODE. Specifically, we develop a generalization of the dynamic manifold trivialization for the manifold ODE forward pass. In a similar spirit, we use a local chart to map to Euclidean space, solve an equivalent Euclidean ODE locally, and project back to the manifold using the chart. The full forward pass is given by Algorithm 1 and is visualized Figure 2.

We present two propositions which highlight that this algorithm is guaranteed to converge to the manifold ODE solution. The first shows that $\varphi_z(\mathbf{y})$ solves the manifold differential equation locally and the second shows that we can pick a finite collection of charts such that we can integrate to time $t_e$.

**Prop 5.1** (Correctness). *If there is some $\epsilon > 0$ such that $\mathbf{y}(t) : [\tau, \tau + \epsilon] \to \mathbb{R}^n$ is a solution to $\frac{d\mathbf{y}(t)}{dt} = D_{\varphi_z(\mathbf{y}(t))}\varphi_z^{-1} \circ f(\varphi_z(\mathbf{y}(t)), t)$ with initial condition $\mathbf{y}(\tau) = \varphi_z^{-1}(z)$, then $\mathbf{z}(t) = \varphi_z(\mathbf{y}(t))$ is a valid solution to Equation 1 on $[\tau, \tau + \epsilon]$.*

**Prop 5.2** (Convergence). *There exists a finite collection of charts $\{\varphi_i\}_{i=1}^k$ s.t. $\mathbf{z}([t_s, t_e]) \subseteq \bigcup\limits_{i=1}^k \text{im}\varphi_i$.*

Proofs of these propositions are given in Appendix A.1. Note that this forward integration is implicit, indicating the connections between [21, 20] and [33]. We realize this construction by finding a principled way to pick charts and incorporate this method into neural networks by developing a backward gradient computation.

We can intuitively visualize this dynamic chart forward pass as a sequence of $k$ Neural ODE solvers $\{\text{ODE}_i\}_{i\in[k]}$ with chart transitions $\varphi_{i_2}^{-1} \circ \varphi_{i_1}$ connecting them. Here, we see how the smooth chart transition property comes into play, as passing between charts is the necessary step in constructing a global solution. In addition, this construction provides a *chart-based backpropagation*. Under this dynamic chart forward pass, we can view a Neural Manifold ODE block as the following composition of Neural ODE blocks and chart transitions:

$$\text{MODE} = \varphi_k \circ \text{ODE}_k \circ (\varphi_k^{-1} \circ \varphi_{k-1}) \circ \cdots \circ (\varphi_2^{-1} \circ \varphi_1) \circ \text{ODE}_1 \circ \varphi_1^{-1} \tag{3}$$

This allows for gradient computation through backpropagation. We may differentiate through the Neural ODE blocks by the Euclidean adjoint method [37, 4].

To finalize this method, we give a strategy for picking these charts for Riemannian manifolds. As previously mentioned, the exponential map serves as a local diffeomorphism from the tangent space (which can be identified with $\mathbb{R}^n$) and the manifold, so it acts as a chart. Similar to [12], at each point

---

Note that we do not lose generality since all manifolds can be given a Riemannian metric.

there exists a radius $r_x$ such that $\exp_x$ is a diffeomorphism when restricted to $B_{r_x} := \{v \in T_x\mathcal{M} : \|v\|_\rho < r_x\}$. With this information, we can solve the equivalent ODE with solution $\mathbf{y}$ until we reach a point $y > (1 - \epsilon)\|r_x\|_\rho$, at which point we switch to the exponential map chart defined around $\exp_x(y)$. In practice, we use the $\exp$ map of the relevant manifold as our local diffeomorphisms. Complete details are provided in Appendix D.

Our dynamic chart method is a significant advancement over previous implicit methods since we can easily construct charts as we integrate. Furthermore, it provides many practical improvements over vanilla Neural Manifold ODEs. Specifically, we

(i) *can perform faster evaluations.* The aforementioned single step algorithms rely on repeated use of the Lie and Riemannian exponential maps. These are expensive to compute and our method can sidestep this expensive evaluation. In particular, the cost is shifted to the derivative of the chart, but by defining dynamics $g$ on the tangent space directly, we can avoid this computation. We use this for our hyperbolic space construction, where we simply solve $\frac{d\mathbf{y}(t)}{dt} = g(\mathbf{y}(t), t)$.

(ii) *avoid catastrophic gradient instability.* If the dimension of $\mathcal{M}$ is less than the dimension of the ambient space, then the tangent spaces are of measure $0$. This means that the induced error from the ODE solver will cause our gradients to leave their domain, resulting in a catastrophic failure. However, since the errors in Neural ODE blocks do not cause the gradient to leave their domain and as our charts induce only a precision error, our dynamic chart method avoids this trap.

(iii) *access a wider array of ODE advancements.* While substantial work has been done for manifold ODE solvers, the vast majority of cutting edge ODE solvers are still restricted to Euclidean space. Our dynamic chart method can make full use of these advanced solvers in the Neural ODE blocks. Additionally, Neural ODE improvements such as [10, 23] can be directly integrated without additional manifold constructions.

## 6 Manifold Continuous Normalizing Flows

With our dynamic chart Neural Manifold ODE, we can construct a Manifold Continuous Normalizing Flow (MCNF). The value can be integrated directly through the ODE, so all that needs to be given is the change in log probability. Here, we can invoke continuous change of variables [4] on the Neural ODE block and can use the smooth chart transition property (which guarantees that the charts are diffeomorphisms) to calculate the final change in probability as:

$$\log p(\text{MODE}) = \log \pi - \sum_{i=1}^{k} \left( \log \det |D\varphi_i| + \log \det |D\varphi_i^{-1}| + \int \text{tr}(D\varphi_i^{-1} \circ f) \right) \quad (4)$$

Note that we drop derivative and integration arguments for brevity. A full expression is given in Appendix B.

For our purposes, the last required computation is the determinant of the $D_v \exp_x$. We find that these can be evaluated analytically, as shown in the cases of the hyperboloid and sphere [36, 40].

Since the MCNF requires only the local dynamics (which are in practice parameterized by the exponential map), this means that the construction *generalizes to arbitrary manifolds*. Furthermore, we *avoid diffeomorphism issues*, such as in the case of two antipodal points on the sphere, simply by restricting our chart domains to never include these conjugate points.

## 7 Experiments

To test our MCNF models, we run density estimation and variational inference experiments. Though our method is general, we take $\mathcal{M}$ to be two spaces of particular interest: hyperbolic space $\mathbb{H}^n$ and the sphere $\mathbb{S}^n$. Appendix B concretely details the computation of MCNF in these spaces. Full experimental details can be found in Appendix C.

### 7.1 Density Estimation

Table 1: MNIST and Omniglot average negative test log likelihood (lower is better) and standard deviation over five trials for varying dimensions.

| | | MNIST | | | Omniglot | | |
|---|---|---|---|---|---|---|---|
| | | 2 | 4 | 6 | 2 | 4 | 6 |
| Euclidean | VAE[27] | $143.06 \pm .3$ | $117.57 \pm .5$ | $102.13 \pm .2$ | $154.31 \pm .5$ | $143.37 \pm .2$ | $138.65 \pm .1$ |
| | RealNVP [8] | $142.09 \pm .7$ | $116.32 \pm .7$ | $100.95 \pm .1$ | $153.93 \pm .5$ | $142.98 \pm .3$ | $\mathbf{137.21} \pm .1$ |
| | CNF [19] | $141.16 \pm .4$ | $116.28 \pm .5$ | $100.64 \pm .1$ | $154.05 \pm .2$ | $143.11 \pm .4$ | $137.32 \pm .7$ |
| Hyperbolic | HVAE[34] | $140.04 \pm .9$ | $114.81 \pm .8$ | $100.45 \pm .2$ | $153.97 \pm .3$ | $144.10 \pm .8$ | $138.02 \pm .3$ |
| | TC [2] | $139.58 \pm .4$ | $114.16 \pm .6$ | $100.45 \pm .2$ | $157.11 \pm 2.7$ | $143.05 \pm .5$ | $137.49 \pm .1$ |
| | WHC [2] | $140.46 \pm 1.3$ | $113.78 \pm .3$ | $100.23 \pm .1$ | $158.08 \pm 1.0$ | $143.23 \pm .6$ | $137.64 \pm .1$ |
| | PRNVP [16] | $140.43 \pm 1.8$ | $113.93 \pm .3$ | $100.06 \pm .1$ | $156.71 \pm 1.7$ | $143.00 \pm .3$ | $137.57 \pm .1$ |
| | MCNF (ours) | $\mathbf{138.14} \pm .5$ | $\mathbf{113.47} \pm .3$ | $\mathbf{99.89} \pm .1$ | $\mathbf{152.98} \pm .1$ | $142.99 \pm .3$ | $137.29 \pm .1$ |

We train normalizing flows for estimation of densities in the hyperbolic space $\mathbb{H}^2$ and the sphere $\mathbb{S}^2$, as these spaces induce efficient computations and are easy to visualize. For hyperbolic space, the baselines are Wrapped Hyperboloid Coupling (WHC) [2] and Projected NVP (PRNVP), which learns RealNVP over the projection of the hyperboloid to Euclidean space [16, 8]. On the sphere, we compare with the recursive construction of [39], with noncompact projection used for the $\mathbb{S}^1$ flow (NCPS). As visualized in Figures 3 and 4, our normalizing flows are able to match complex target densities with significant improvement over the baselines. MCNF is even able to fit discontinuous and multi-modal target densities; baseline methods cannot fit these cases as they struggle with reducing probability mass in areas of low target density.

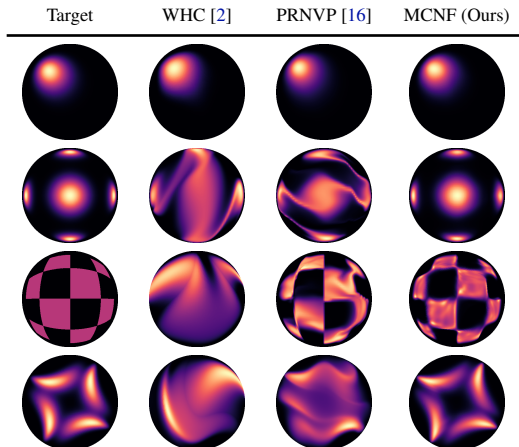

Figure 3: Density estimation on the hyperboloid $\mathbb{H}^2$, which is projected to the Poincaré Ball for visualization.

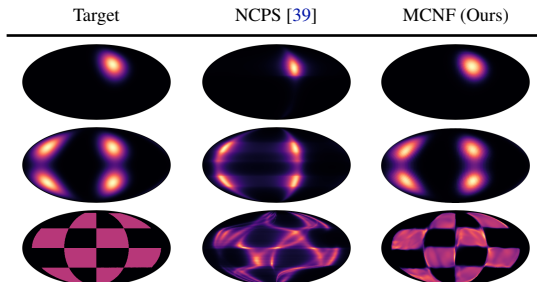

Figure 4: Density estimation on the sphere $\mathbb{S}^2$, which is projected to two dimensions by the Mollweide projection.

## 7.2 Variational Inference

We train a hyperbolic VAE [34] and Euclidean VAE [27] for variational inference on Binarized Omniglot [28] and Binarized MNIST [30]. Both of these datasets have been found to empirically benefit from hyperbolic embeddings in prior work [36, 34, 2, 24, 40]. We compare different flow layers in the latent space of the VAEs. For the Euclidean VAE, we compare with RealNVP [8] and CNF [4, 19]. Along with the two hyperbolic baselines used for density estimation, we also compare against the Tangent Coupling (TC) model in [2]. As shown in Table 1, our continuous flow regime is more expressive and learns better than all hyperbolic baselines. In low dimensions, along with the other hyperbolic models, our approach tends to outperform Euclidean models on Omniglot and MNIST. However, in high dimension the hyperbolic models do not reap as much benefit; even the baseline HVAE does not consistently outperform the Euclidean VAE.

## 8    Conclusion

We have presented Neural Manifold ODEs, which allow for the construction of continuous time manifold neural networks. In particular, we introduce the relevant theory for defining "pure" Neural Manifold ODEs and augment it with our dynamic chart method. This resolves numerical and computational cost issues while allowing for better integration with modern Neural ODE theory. With this framework of continuous manifold dynamics, we develop Manifold Continuous Normalizing Flows. Empirical evaluation of our flows shows that they outperform existing manifold normalizing flow baselines on density estimation and variational inference tasks. Most importantly, our method is completely general as it does not require anything beyond local manifold structure.

We hope that our work paves the way for future development of manifold-based deep learning. In particular, we anticipate that our general framework will lead to other continuous generalizations of manifold neural networks. Furthermore, we expect to see application of our Manifold Continuous Normalizing Flows for topologically nontrivial data in lattice quantum field theory, motion estimation, and protein structure prediction.

## 9    Broader Impact

As mentioned in the introduction, our method has applications to physics, robotics, and biology. While there are ethical and social concerns with parts of these fields, our work is too theoretical for us to say for sure what the final impact will be. For deep generative models, there are overarching concerns with generating fake data for malicious use (e.g. deepfake impersonations). However, our work is more concerned with accurately modelling data topology rather than generating hyper-realistic vision or audio samples, so we do not expect there to be any negative consequence in this area.

## 10    Acknowledgements

We would like to acknowledge Prof. Austin Benson and Junteng Jia for their insightful comments and suggestions. In addition, we would like to thank Facebook AI for funding equipment that made this work possible. This work is also supported by the grant NSF-2008102: RI: Small: Reliable Machine Learning in Hyperbolic Spaces. We would like to thank Joey Bose for providing access to his prerelease code, without which the comparison shown in the paper would not be possible.

## Footnotes

\* indicates equal contribution

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
