[Supplementary Material]

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

# A  Proofs of Propositions

## A.1  Dynamic Chart Method

**Prop A.1** (Correctness). *If $\mathbf{y}(t) : [\tau, \tau + \epsilon] \to \mathbb{R}^n$ is a solution to $\frac{d\mathbf{y}(t)}{dt} = D_{\varphi_z(\mathbf{y}(t))}\varphi_z^{-1} \circ f(\varphi_z(\mathbf{y}(t)), t)$ with initial condition $\mathbf{y}(\tau) = \varphi_z^{-1}(z)$, then $\mathbf{z}(t) = \varphi_z(\mathbf{y}(t))$ is a valid solution to Equation 1 on $[\tau, \tau + \epsilon]$.*

*Proof.* We see that if $\mathbf{z}(t) = \varphi_z(\mathbf{y}(t))$ then for all $t' \in [\tau, \tau + \epsilon]$

$$\frac{d\mathbf{z}(t')}{dt} = D_{\mathbf{y}(t')}\varphi_z \circ \frac{d\mathbf{y}(t')}{dt} \tag{5}$$

$$= D_{\mathbf{y}(t')}\varphi_z \circ D_{\varphi_z(\mathbf{y}(t'))}\varphi_z^{-1} \circ f(\varphi_z(\mathbf{y}(t')), t') \tag{6}$$

$$= f(\varphi_z(\mathbf{y}(t')), t') \tag{7}$$

$$= f(\mathbf{z}(t'), t') \tag{8}$$

$\square$

**Prop A.2** (Convergence). *There exists a finite collection of charts $\{\varphi_i\}_{i=1}^k$ s.t. $\mathbf{z}([t_s, t_e]) \subseteq \bigcup_{i=1}^{k} \mathrm{im}\varphi_i$.*

*Proof.* Around each point $z \in \mathbf{z}([t_s, t_e])$ pick a chart $\varphi_z$. Then note that $\{\varphi_z\}_{z \in \mathbf{z}([t_s, t_e])}$ satisfies $\mathbf{z}([t_s, t_e]) \subseteq \bigcup_z \mathrm{im}\varphi_z$. But, the curve is compact since $[t_s, t_e]$ is compact (and $\mathbf{z}$ is assumed to be continuous) so we can take a finite subset $\{\varphi_i\}_{i=1}^k$ that covers the curve. $\square$

## A.2  Derivation of Gradient for Adjoint State

In this section we prove Theorem 4.1. The proof follows from the analogous one in [4], though we replace certain operations with their manifold counterparts.

**Theorem A.3.** *Suppose we have some manifold ODE as given in Equation 1 and we define some loss function $L : \mathcal{M} \to \mathbb{R}$. Suppose that there is an embedding of $\mathcal{M}$ in some Euclidean space $\mathbb{R}^d$ and we identify $T_x\mathcal{M}$ with an $n$-dimensional subspace of $\mathbb{R}^d$. If we define the adjoint state to be $\mathbf{a}(t) = D_{\mathbf{z}(t)}L$, then the adjoint satisfies*

$$\frac{d\mathbf{a}(t)}{dt} = -\mathbf{a}(t)D_{\mathbf{z}(t)}f(\mathbf{z}(t), t) \tag{9}$$

*Proof.* Consider the first order approximation of $\mathbf{z}(t + \epsilon)$. Since we are embedding $T_x\mathcal{M} \subseteq \mathbb{R}^d$, then under standard $\mathbb{R}^d$ computations we have that

$$\mathbf{z}(t + \epsilon) = \mathbf{z}(t) + \epsilon f(\mathbf{z}(t), t) + \mathcal{O}(\epsilon^2) \tag{10}$$

We set $T_\epsilon(\mathbf{z}(t), t) := \mathbf{z}(t + \epsilon)$. As in the original adjoint method derivation [4], we utilize the chain rule

$$D_{\mathbf{z}(t)}L = D_{\mathbf{z}(t+\epsilon)}L \circ D_{\mathbf{z}(t)}T_\epsilon(\mathbf{z}(t), t) \quad \text{or} \quad \mathbf{a}(t) = \mathbf{a}(t + \epsilon)D_{\mathbf{z}(t)}T_\epsilon(\mathbf{z}(t), t) \tag{11}$$

Using these, we get that

$$\frac{d\mathbf{a}(t)}{dt} = \lim_{\epsilon \to 0+} \frac{\mathbf{a}(t + \epsilon) - \mathbf{a}(t)}{\epsilon} \tag{12}$$

$$= \lim_{\epsilon \to 0+} \frac{\mathbf{a}(t + \epsilon) - \mathbf{a}(t + \epsilon)D_{\mathbf{z}(t)}T_\epsilon(\mathbf{z}(t), t)}{\epsilon} \quad \text{(by Equation 11)} \tag{13}$$

$$= \lim_{\epsilon \to 0+} \frac{\mathbf{a}(t + \epsilon) - \mathbf{a}(t + \epsilon)D_{\mathbf{z}(t)}(\mathbf{z}(t) + \epsilon f(\mathbf{z}(t), t) + \mathcal{O}(\epsilon^2))}{\epsilon} \quad \text{(by Equation 10)} \tag{14}$$

$$= \lim_{\epsilon \to 0+} \frac{-\epsilon \mathbf{a}(t + \epsilon)D_{\mathbf{z}(t)}f(\mathbf{z}(t), t) + O(\epsilon^2)}{\epsilon} \tag{15}$$

$$= -\mathbf{a}(t)D_{\mathbf{z}(t)}f(\mathbf{z}(t), t) \tag{16}$$

$$\square$$

## B   Computation of Manifold Continuous Normalizing Flows

### B.1   Background

In Euclidean space, a normalizing flow $f$ is a diffeomorphism $f : \mathbb{R}^n \to \mathbb{R}^n$ that maps a base probability distribution into a more complex probability distribution. Suppose $z \sim \pi$ is a sample from the simple distribution. By the change of variables formula, the target density of an $x$ in terms of $p$ (the complex distribution) is

$$\log p(x) = \log \pi(z) - \log \det \left| \frac{\partial f^{-1}}{\partial z} \right| \tag{17}$$

There exist a variety of functions $f$ which constrain the log Jacobian determinant to be computationally tractable. An important one is the Continuous Normalizing Flow (CNF). CNFs construct $f$ to be the solution of an ODE [4, 19]. Explicitly, let $t_0, t_1$ be starting and ending times with $t_0 < t_1$, and consider the ordinary differential equation $\frac{d\mathbf{z}(t)}{dt} = g(\mathbf{z}(t), t; \theta)$, where $\theta$ parameterizes the dynamics $g$. For a sample from the base distribution $z \sim \pi$, solving this ODE with initial condition $\mathbf{z}(t_0) = z$ gives the sample from the target distribution $x = \mathbf{z}(t_1)$. The change in the log density given by this model satisfies an ordinary differential equation called the instantaneous change of variables formula:

$$\frac{d \log p(\mathbf{z}(t))}{dt} = -\text{tr}\left( D_{\mathbf{z}(t)}g \right) \tag{18}$$

We can therefore quantify the final probability as

$$\log p(\mathbf{z}(t_1)) = \log p(\mathbf{z}(t_0)) - \int_{t_0}^{t_1} \text{tr}\left( D_{\mathbf{z}(t)}g \right) \, dt. \tag{19}$$

### B.2   Manifold Continuous Normalizing Flows

For our MCNF we split up the original time $[t_s, t_e]$ into $[t_i, t_{i+1}]$ for $i \in [k]$ where $t_s = t_1 < t_2 < \cdots < t_k < t_{k+1} = t_e$. From our curve $\mathbf{z}$ we can select $z_i = \mathbf{z}(t_i)$, and we have charts $\varphi_i : U_i \to V_i$ s.t. $z_i, z_{i+1} \in V_i$. If our dynamics are determined by $\frac{d\mathbf{z}(t)}{dt} = f(\mathbf{z}(t), t; \theta)$ then this locally takes the form $\varphi_i(\widehat{f}_i(\varphi_i^{-1}(\mathbf{z}(t), t; \theta)))$, in which $\widehat{f}_i$ is a CNF. The update after passing through a chart $\varphi_i$ and integrating is given by

$$\log p(z_{i+1}) = \log \pi(z_i) - \left( \log \left| \det D_{\widehat{f}_i(\varphi^{-1}(z_i))}\varphi_i \right| + \int_{t_i}^{t_{i+1}} \text{tr}(D_{\varphi_i^{-1}(z_i)}\widehat{f}_i)dt + \log \left| \det D_{z_i}\varphi_i^{-1} \right| \right)$$

where $\log|\det D\varphi_i|$ is a shorthand for the Riemannian probability update induced by the chart. Note that in general this is not the determinant (for instance when the map goes from elements in $\mathbb{R}^n$ to $\mathbb{R}^d$ where $d > n$). In practice it ends up being quite similar.

(a) $\mathcal{G}(\mu_0, I)$       (b) $\mathrm{vMF}(\mu_0, 1)$

Figure 5: Base distributions used for flow models. In $\mathbb{H}^2$, $\mu_0 = (1, 0, 0)$ and in $\mathbb{S}^2$, $\mu_0 = (-1, 0, 0)$. (a) is on $\mathbb{H}^2$ as visualized on the Poincaré ball. (b) is on $\mathbb{S}^2$ as visualized by the Mollweide projection.

We can consider our manifold ODE as a composition of these updates. Therefore, we have that

$$
\begin{aligned}
\log & p(f(z)) = \\
& \log \pi(z) - \sum_{i=1}^{k} \left( \log \left| \det D_{\widehat{f}_i(\varphi^{-1}(z_i))} \varphi_i \right| + \int_{t_i}^{t_{i+1}} \mathrm{tr}(D_{\varphi_i^{-1}(z_i)} \widehat{f}_i) dt + \log \left| \det D_{z_i} \varphi_i^{-1} \right| \right)
\end{aligned}
\tag{20}
$$

For our cases we will be setting $\varphi_i = \exp_{z_i}$.

### B.3 Base Distributions

**Hyperbolic Space.** We will use the hyperbolic wrapped normal distribution $\mathcal{G}(\mu, \Sigma)$ where $\mu \in \mathcal{M}$ and $\Sigma \in \mathbb{R}^{n \times n}$ where $\mathcal{M}$ has dimension $n$ [36].

1. **Sampling.** To sample a $m \in \mathcal{M}$, perform the following steps. A priori, set some $\mu_0 \in \mathcal{M}$. First, sample $v \in \mathcal{N}(\mu_0, \Sigma) \in T_{\mu_0}\mathcal{M}$. Then parallel transport this vector to the mean $\mu$ i.e. $u = \mathrm{PT}_{\mu_0 \to \mu}(v)$. Lastly, project to the manifold using the exponential map $m = \exp_\mu(u)$.

2. **Probability Density.** The probability density can be found through the composition of the parallel transport map and exponential map. Specifically, we have that

$$
\log p(x) = \log p(v) - \log \left| \det D_u \exp_\mu(u) \right| - \log \left| \det D_v \mathrm{PT}_{\mu_0 \to \mu}(v) \right| \tag{21}
$$

**Spherical Space.** We could possibly perform a relatively similar wrapped normal distribution [40]. However, we see that this is theoretically flawed since parallel transport between two conjugate points is not well defined.

Instead, we will use the von Mises-Fisher distribution, a distribution on the $(n-1)$-sphere in $\mathbb{R}^n$ originally derived for use in directional statistics [15]. We denote this distribution as $\mathrm{vMF}(\mu, \kappa)$ where $\mu \in \mathbb{R}^n$ is treated as an element of $\mathbb{S}^{n-1}$ via the canonical embedding, and $\kappa \in \mathbb{R}_{\geq 0}$. Note the following about the von Mises-Fisher distribution:

1. **Sampling.** The von Mises-Fisher can be sampled from with efficient methods [42, 7].

2. **Probability Density.** From [15, 6], we know that the density is given by

$$
p(z) = \mathcal{C}_n(\kappa) \exp(\kappa \mu^T z) \tag{22}
$$

$$
\mathcal{C}_n(\kappa) = \frac{\kappa^{n/2-1}}{(2\pi)^{n/2} \mathcal{I}_{n/2-1}(\kappa)} \tag{23}
$$

Note that $||\mu||^2 = 1$, $\mathcal{C}_n(\kappa)$ is the normalizing constant, and that $\mathcal{I}_v$ denotes the modified Bessel function of the first kind of order $v$.

These baseline probability distributions are visualized in Figure 5.

Table 2: Formulas for basic operations in hyperbolic space $\mathbb{H}^n$.

| | |
|---|---|
| Manifold | $\mathbb{H}^n = \{x \in \mathbb{R}^{n+1} : \|x\|_{\mathcal{L}} = -1, x_0 > 0\}$ |
| Tangent space | $T_x\mathbb{H}^n = \{v \in \mathbb{R}^{n+1} : \langle x, y \rangle_{\mathcal{L}} = 0\}$ |
| Exponential map | $\exp_x(v) = \cosh(\|v\|_{\mathcal{L}})x + \sinh(\|v\|_{\mathcal{L}})\frac{v}{\|v\|_{\mathcal{L}}}$ |
| Logarithmic map | $\log_x(y) = \frac{\text{arccosh}(\langle x,y\rangle_{\mathcal{L}})}{\sinh(\text{arccosh}(\langle x,y\rangle_{\mathcal{L}}))}(y - \langle x,y\rangle_{\mathcal{L}} x)$ |
| Parallel transport | $\text{PT}_{x \to y}(v) = v - \frac{\langle y,v\rangle_{\mathcal{L}}}{1 + \langle x,y\rangle_{\mathcal{L}}}(x + y)$ |
| Tangent projection | $\text{proj}_x(u) = u + \langle x, u\rangle_{\mathcal{L}} x$ |

## B.4 Hyperbolic Space

### B.4.1 Analytic Derivations

For hyperbolic space, we will work with the hyperboloid $\mathbb{H}^n$. The analytic values of the operations are given in Table 2. Recall that the *Lorentz Inner Product and Norm* are given by

$$\langle x, u\rangle_{\mathcal{L}} := -x_0 u_0 + x_1 u_1 + \dots x_n u_n \qquad \|x\|_{\mathcal{L}} = \sqrt{\langle x, x\rangle_{\mathcal{L}}} \qquad (24)$$

In addition, there are many useful identities which appear in our pipeline.

1. **Stereographic Projection.** To visualize $\mathbb{H}^2$ on the Poincaré ball, we use the stereographic projection as explained in [40], which maps a point $(\xi, x^T) \in \mathbb{H}^2 \subseteq \mathbb{R}^3$ to a the point $x/(1 + \xi) \subseteq \mathbb{R}^2$ on the Poincaré ball.

2. **Log Determinants.** The log determinant of the derivative of the exponential map is given by [36, 40]:

$$\log |\det D_v \exp_x| = (n - 1)\left[\log \sinh(\|v\|_{\mathcal{L}}) - \log \|v\|_{\mathcal{L}}\right] \qquad (25)$$

   The log determinant of the derivative of the log map is the negation of the above by the inverse function theorem, and the log determinant of the derivative of parallel transport is $0$.

### B.4.2 Numerical Stability

In order to ensure numerical stability, we examine several operations which are inherently numerically unstable and present solutions

1. **Arccosh.** Arccosh has a domain of $(-\infty, -1) \cup (1, +\infty)$. In practice, we are concerned with the positive case, although the negative case can be similarly handled. Due to numerical instability a value $1 + \epsilon$ may be realized as $1$ in our floating point system. To compensate, we clamp the minimum value to be $1 + \epsilon_0$ for a small fixed $\epsilon_0$.

2. **Sinh Division.** In the exponential and logarithmic maps, there exist terms of the form $\frac{\sinh(x)}{x}$. When $|x| < \epsilon$ for some small $\epsilon$, this is numerically unstable. We special case this (and the derivative) for stability by explicitly deriving the limit value of $x \to 0$ for these cases.

## B.5 Spherical Space

### B.5.1 Analytic Derivations

For spherical space, we work with the sphere $\mathbb{S}^n$. The analytic values are given in Table 3. Norms and inner products are assumed to be the Euclidean $\ell^2$ values.

Some useful identities are

Table 3: Formulas for basic operations on the sphere $\mathbb{S}^n$.

| | |
|---|---|
| Manifold | $\mathbb{S}^n = \{x \in \mathbb{R}^{n+1} : \|x\| = 1\}$ |
| Tangent Space | $T_x\mathbb{S}^n = \{v \in \mathbb{R}^{n+1} : \langle x, v \rangle = 0\}$ |
| Exponential map | $\exp_x(y) = \cos(\|v\|)x + \sin(\|v\|)\frac{v}{\|v\|}$ |
| Logarithmic map | $\log_x(y) = \frac{\arccos(\langle x,y\rangle)}{\sin(\arccos(\langle x,y\rangle))}(y - \langle x, y \rangle x)$ |
| Parallel transport | $\mathrm{PT}_{x \to y}(v) = v - \frac{\langle y,v\rangle}{1 + \langle x,y\rangle}(x + y)$ |
| Tangent projection | $\mathrm{proj}_x(u) = u - \langle x, u \rangle x$ |

1. **Mollweide Projection.** To visualize $\mathbb{S}^2$, we use the Mollweide projection that is used in cartography. For latitude $\theta$ and longitude $\varphi$, the sphere is projected to coordinates $(x, y)$ by (where $\beta$ is a variable solved for by the given equation) [29]:

$$2\beta + \sin(2\beta) = \pi \sin(\varphi), \quad x = \frac{2\sqrt{2}}{\pi}\varphi\cos(\beta), \quad y = \sqrt{2}\sin(\beta) \tag{26}$$

2. **Log Determinants.** The log determinant for the exponential map of the Sphere is given by

$$\log|\det D_v \exp_x| = (n - 1)\left[\log\sin(\|v\|) - \log\|v\|\right] \tag{27}$$

The log determinant of the derivative of the log map is the negation of the above and the log determinant of the derivative of parallel transport is $0$.

### B.5.2 Numerical Stability

In order to ensure numerical stability, we note that several functions are inherently numerically unstable

1. **Sine division.** In the exp and log maps, there are values of the form $\frac{\sin x}{x}$. Note that this is numerically unstable when $|x| < \epsilon$. We special case this (and its derivative) to allow for better propagation.

2. **Log Derivative.** We find that we need an explicit derivation of $D_x \log y$ for our Manifold ODE on the Sphere (see B.7). Note that this value can be computed using backpropagation, but we derive it explicitly instead due to numerical instability of the higher order derivatives of some of our functions.

**Lemma.** *For $x, y \in \mathcal{S}^n$, and $r = x^T y$, if $|r| \neq 1$, then*

$$D_y \log_x(y) = \left(\frac{r\arccos(r)}{(1 - r^2)^{3/2}} - \frac{1}{1 - r^2}\right)(y - rx)x^T + \frac{\arccos(r)}{\sin(\arccos(r))}\left(I - xx^T\right) \tag{28}$$

*The limit of $D_y \log_x(y)$ as $r = x^T y \to 1$ is $I - xx^T$.*

*Proof.* We differentiate the equation of the logarithmic map as given in Table 3. First, suppose that $|r| \neq 1$. By the product rule we have,

$$D_y \log_x(y) = D_y\left(\frac{\arccos(r)}{\sin(\arccos(r))}\right)(y - rx) + \frac{\arccos(r)}{\sin(\arccos(r))}D_y(y - rx).$$

The summand on the right is given by

$$\frac{\arccos(r)}{\sin(\arccos(r))}(I - xx^T).$$

To compute the left summand, we use the chain rule and differentiate by $r$

$$D_y \left( \frac{\arccos(r)}{\sin(\arccos(r))} \right) (y - rx) = \frac{\partial}{\partial r} \left( \frac{\arccos(r)}{\sin(\arccos(r))} \right) (y - rx)x^T$$

$$= \left( \frac{r \arccos(r)}{(1 - r^2)^{3/2}} - \frac{1}{1 - r^2} \right) (y - rx)x^T$$

To check that the limit as $x^T y \to 1$ is $I - xx^T$, it is enough to compute three separate limits that are all finite. It is clear that

$$\lim_{y \to x} (y - rx)x^T = 0.$$

Since the limit of the other term in the left summand can be shown to be finite, this means that the left summand is zero in the limit. For the right summand, the only term that depends on $y$ has a limit

$$\lim_{r \to 1} \frac{\arccos(r)}{\sin(\arccos(r))} = \lim_{r \to 1} \frac{\arccos(r)}{\sqrt{1 - r^2}} = 1$$

where the final equality can be seen by L'Hopital's rule. Thus, the entire limit is $I - xx^T$. $\square$

## B.6  Backpropagation

To update the parameters of an MCNF, we need to differentiate $\log p(\text{MODE})$ with respect to $\theta$, so we need to differentiate each of the summands in (4) with respect to $\theta$. Differentiating through neural ODE blocks is done with the Euclidean adjoint method [37, 4, 19], which, for a loss $L$ depending on the solution $\mathbf{y} : [t_s, t_e] \to \mathbb{R}^n$ to a differential equation with dynamics $g(\mathbf{y}(t), t; \theta)$, gives that

$$\frac{\partial L}{\partial \theta} = - \int_{t_e}^{t_s} \frac{\partial L}{\partial \mathbf{y}(t)} \frac{\partial g(\mathbf{y}(t), t; \theta)}{\partial \theta} \, dt \tag{29}$$

Differentiating the dynamics is done with standard backpropagation. The adjoint state $\frac{\partial L}{\partial \mathbf{y}(t)}$ is computed by the solution of the adjoint differential equation (2) for Euclidean space, with initial condition $\frac{\partial L}{\partial \mathbf{y}(t_e)}$. The derivative of the loss $\frac{\partial L}{\partial \mathbf{y}(t_e)}$ can be computed directly. For MCNF, $L$ is taken to be the negative log likelihood.

For the hyperbolic and spherical cases, the log determinant terms take simple forms (as in equations 25 and 27), and are thus easy to differentiate through. Moreover, for the hyperbolic VAE models, we train by maximizing the evidence lower bound (ELBO) on the log likelihood, so that differentiation is done with a reparameterization as in [36].

## B.7  Designing Neural Networks

### B.7.1  Construction

In general we construct the dynamics $f(\mathbf{z}(t), t) \in T_{\mathbf{z}(t)}\mathcal{M}$ as follows. Suppose $\mathcal{M}$ is embedded in some $\mathbb{R}^d$. Then we construct $f$ to be a neural network with input of dimension $d + 1$. The first $d$ values are the manifold input and the last value is the time. The output of the neural network will be some vector in $\mathbb{R}^d$. To finalize, we project onto the tangent space using the linear projection $\text{proj}_{\mathbf{z}(t)}$.

**Hyperbolic Space.** Since hyperbolic space $\mathbb{H}^n$ is diffeomorphic to Euclidean space, we can parameterize all manifold dynamics with a corresponding Euclidean dynamic with an $\exp_{\mu_0}$, where $\mu_0$ is the point $(1, 0, \ldots, 0) \in \mathbb{R}^{n+1}$. In general since we only require one chart, our Manifold ODE consists of $\exp_0 \circ \text{ODE} \circ \log_0$, which means that we can model our full dynamics only in the tangent space (not on the manifold). By picking our basis, we can represent elements of $T_0\mathcal{M}$ as element of $\mathbb{R}^n$. For the ODE block, we parameterize using a neural network $f$ which takes in an input of dimension $n + 1$ (which is a tangent space element and time) and outputs an element of dimension $n$.

**Spherical Space.**

For the spherical case, we use the default construction (with projection), as there is no global diffeomorphism. Note that when passing from the manifold to tangent space dynamics, we require $D_y \log_x$. We also must invoke a radius of injectivity, as opposed to hyperbolic space. This is $\pi$ for all points.

### B.7.2 Existence of a Solution

We construct our networks in such a way that the Picard-Lindelöf theorem holds. Our dynamics are given by $D_z \varphi \circ f$ where $\varphi$ is a chart and $f$ is a neural network. These are well behaved since 1) the neural network dynamics are well behaved using tanh and other Lipchitz nonlinearities and 2) the chart is well behaved since we can bound the domain to be compact.

## C  Experimental Details

### C.1  Data

In our code release, we will include functions to generate samples from the target densities that are used for density estimation in section 7.1.

**Hyperbolic Density Estimation** We detail the hyperbolic densities in each row of Figure 3.

1. The hyperbolic density in the first row of Figure 3 is a wrapped normal distribution with mean $(-1, 1)$ and covariance $\frac{3}{4}I$.

2. The second density is built from a mixture of 5 Euclidean gaussians, with means $(3, 0), (-3, 0), (0, 3), (0, -3)$, and $(0, 0)$, and covariance $\frac{1}{2}I$. The resulting hyperbolic density is obtained by viewing $\mathbb{R}^2$ as the tangent space $\mathcal{T}_0 \mathcal{M}$, and then projecting the Euclidean density to the Hyperboloid by $\exp_0$.

3. The third density is a projection onto the hyperboloid of a uniform checkerboard density in $\mathcal{T}_0 \mathcal{M}$. The square in the second row and third column of the checkerboard has its lower-left corner at the origin $(0, 0)$. Each square has side length $1.5$.

4. The fourth density is a mixture of four wrapped normal distributions. Letting $s = 1.3$, $\sigma_1 = .3$ and $\sigma_2 = 1.5$, the wrapped normals are given as:

$$\mathcal{G}\left((0, s, s), \mathrm{diag}(\sigma_1, \sigma_2)\right) \qquad \mathcal{G}\left((0, -s, -s), \mathrm{diag}(\sigma_1, \sigma_2)\right)$$
$$\mathcal{G}\left((0, -s, s), \mathrm{diag}(\sigma_2, \sigma_1)\right) \qquad \mathcal{G}\left((0, s, -s), \mathrm{diag}(\sigma_2, \sigma_1)\right)$$

**Spherical Density Estimation** Details are given about the densities that were learned in each row of Figure 4.

1. The density in the first row of Figure 4 is a wrapped normal distribution with mean $\frac{1}{\sqrt{3}}(-1, -1, -1)$ and covariance $\frac{3}{10}I$.

2. The second density is built from a mixture of 4 wrapped normals, with means $\frac{1}{\sqrt{3}}(1, 1, 1), \frac{1}{\sqrt{3}}(-1, -1, -1), \frac{1}{\sqrt{3}}(-1, -1, 1)$, and $\frac{1}{\sqrt{3}}(1, 1, -1)$; all components of the mixture have covariance $\frac{3}{10}I$.

3. The third density is a uniform checkerboard density in spherical coordinates $(\varphi, \theta) \in [0, 2\pi] \times [0, \pi]$. The rectangle in the second row and third column of the checkerboard has its lower-left corner at $(\pi, \pi/2)$. The side length of each rectangle in the $\varphi$-axis is $\pi/2 - 0.2$, and the side length in the $\theta$-axis is $\pi/4 - 0.1$.

**Variational Inference** For variational inference, we dynamically binarize the MNIST and Omniglot images with the same procedure as given in [40]. We resize the Omniglot images to $28 \times 28$, the same size as the MNIST images.

### C.2  Density Estimation

In section 7.1 we train on batches of 200 samples from the target density (or batches of size 100 for the discrete spherical normalizing flows [39]). Our MCNF models and the hyperbolic baselines use at most 1,000,000 samples, with early stopping as needed—the hyperbolic baselines sometimes diverge when training for too many batches. As suggested in [39], we find that the spherical baseline does indeed needed more samples than this (at least 5,000,000), so we allow it to train until the density converges. Although we do not investigate sample efficiency in detail, we find that our MCNF is able to achieve better results than the spherical baseline with, frequently, over an order of magnitude

fewer samples than the spherical baseline. Note that all methods use the Adam optimizer [25]. For our MCNF, our dynamics are given by a neural network of hidden dimension 32 and 4 linear layers with tanh activation; for each integration we use a Runge-Kutta 4 solver.

**Hyperbolic Normalizing Flows** For the hyperbolic discrete normalizing flows, we train with 4 hidden blocks, hidden flow dimension of 32, and tanh activations. The prior distributions used are given in section B.3 and target distributions are given in section C.1.

**Spherical Normalizing Flows** For the discrete spherical normalizing flows from [38], we use the recursive flow for $\mathbb{S}^2$. In designing this flow, we use the non-compact projection (NCP) flow for $\mathbb{S}^1$ and the autoregressive spline flow from [11] for the interval $[-1, 1]$. To increase expressiveness for the $\mathbb{S}^1$ flow, we consider learning a convex combination of NCP flows over the circle. Let the number of components in this combination be $n$. To increase the expressiveness of the spline flow over $[-1, 1]$, we increase the number of segments. Let the number of segments be $k$. We tuned $n$ and $k$ for each of the 3 spherical densities to yield the best results. Note that the best $n$ and $k$ frequently ended up being fairly small for the more simple densities, since having an overly expressive model for simple densities ended up being hard to train and produced undesirable artifacts.

The prior distributions used for all target distributions are given in section B.3. For the first spherical target distribution in section C.1, we use $k = 2$ and $n = 1$. For the second spherical target distribution we use $k = 6$ and $n = 2$. For the final spherical target distribution we use $k = 32$ and $n = 12$.

## C.3 Variational Inference

In each model for variational inference, we train with a hyperbolic or Euclidean VAE. Following [2], in both cases, the mean and variance encoders are taken to be one layer neural networks with a hidden dimension of 600. The $\mathrm{ReLU}$ nonlinearity is used for the hyperbolic VAE and the $\mathrm{LeakyReLU}$ is used for the Euclidean VAE. As is often done, we take the weight matrices of the first linear layer to be shared for the mean and variance [2, 27]. The decoder is taken to have a symmetric architecture, with the input coming from the latent space and the output being a decoded image. We vary the latent dimension from 2 to 6 in our experiments. When we add a discrete normalizing flow, we use 2 hidden blocks and 2 hidden dimensions per block with a hidden layer of size 128 and tanh activations, again following [2]. For continuous flows, we replace this with a neural ODE or MCNF block where the dynamics are parameterized by a two-layer network of hidden size 128 with tanh activations. For numerical integration we use the Runge-Kutta 4 solver.

# D Dynamic Chart Method

Here we elaborate on the dynamic chart method presented in Section 5 of our paper. Specifically, we discuss the significance of dynamic charts and the generality of the multi-chart MCNF (which allows learning of arbitrary densities on manifolds with conjugate points, like $\mathbb{S}^2$).

## D.1 Choosing Charts

**The Benefit of Dynamic Charts.** Note that our dynamic chart trivialization is performed in the main paper simply by splitting the time interval $[t_s, t_e]$ up uniformly into segments of length $\epsilon$ and learning the solution locally via exp-map charts at the anchor points (endpoints of the segments). Such a splitting allows for the *ball of injectivity* (induced by the radius of injectivity) to "move" throughout the training process, so that it always surrounds the locally relevant region (centered at the anchor point). This approach allows for transportation of mass that evades the conjugate point problem (which [16] does not allow for).

This becomes especially clear if we consider the case of conjugate points on a sphere, i.e. the case of antipodal points. Consider a task in which we have a prior with mass surrounding one antipodal point and the target density is concentrated around the other point. A one chart approach would have trouble transporting the mass due to the fact that a fixed exp-map can never have a ball of injectivity (induced by the radius of injectivity $\pi$, in this case) that includes both points throughout training. However, our dynamic approach allows this ball of injectivity to shift throughout the training process and makes correct transportation of mass for such a scenario possible. An experiment testifying to this is given in Appendix D.2 (and Figure 6).

(a) Target

(b) 1 chart

(c) 16 charts

Figure 6: Comparison of MCNF for different numbers of charts. Note that using just 1 chart is not enough to learn the density concentrated at the antipode, while the 16 chart model learns the density well.

**Non-uniform Time-domain Segmenting.** While our approach allows for the ball of injectivity to shift, the dynamic chart method is not limited to a uniform $\epsilon$-segment splitting of the $[t_s, t_e]$ interval. Similar benefits may be derived with an alternative splitting. However, it is not clear what additional benefits a non-uniform splitting might bring without prior knowledge of local manifold topology. Our experiments find that a uniform split works well for the densities we tested.

**Control of Local Dynamics** Note that although the dynamic chart approach makes it possible for the ball of injectivity to move (making it possible to learn general densities), local dynamics may still cause issues. This is because the local dynamics of the ODE may cause the solver to venture to the edges of the ball of injectivity surrounding the current anchor point $x$, thereby causing instability. One may resolve this issue by enforcing a Lipschitz constraint on the solution by explicitly bounding the derivative (e.g. bounding $\frac{d\mathbf{y}(t)}{dt}$). If we call the Lipschitz constant $L \in \mathbb{R}$, the length of chart domain $\epsilon > 0$, and the radius of injectivity $r_x$ at the current anchor point $x$, we would want to enforce the constraint such that $\epsilon L < r_x$, i.e. $L < r_x/\epsilon$. Notice that this is only a local Lipschitz constraint, since it depends on the radius of injectivity at each anchor point, and moreover, on the way the time-domain segmenting is done. Enforcing this constraint, even in a simplistic way (e.g. thresholding $\frac{d\mathbf{y}(t)}{dt}$ at $r_x/\epsilon$), ensures that the dynamics are configured so the local solution stays within the ball of injectivity and instability is avoided. We note that for most of our experiments, we did not worry about this, as our method was already stable, but we include this clarification in the case that it becomes necessary to maintain stability (for e.g. a particularly complicated density on a manifold with conjugate points).

### D.2 Expressivity and Generality of MCNF

To demonstrate the effectiveness of our dynamic chart method in getting around conjugate points and numerical instability around them, we set up a particular density on the sphere for estimation. The target density is a $\text{vMF}((1, 0, 0), 30)$ distribution, which is heavily concentrated around $(1, 0, 0)$, as shown in Figure 6 (a). We still take our base distribution as the vMF centered at $\mu_0 = (-1, 0, 0)$ (see Figure 5), but now we take it to be more concentrated by setting $\kappa = 3$. To learn the target density using this base distribution, our MCNF must learn to move probability density from samples around $(1, 0, 0)$ to areas of high base probability density around the antipodal point $(-1, 0, 0)$.

As shown in Figure 6, MCNF with just one chart is not able to learn a density with high probability at $(1, 0, 0)$, as it is unable to move samples around $(1, 0, 0)$ close enough to the antipodal point. On the other hand, MCNF with 16 charts is able to do so, thus validating our model's ability to get around conjugate points with the dynamic chart method.

# E Generated MNIST Samples

Here, we generate samples from MNIST using our trained hyperbolic MCNF. To do this, we use an MCNF with the same setup as in C.3, except with a slightly larger VAE architecture. We add a linear layer to both the mean and variance networks, add an additional shared linear layer for both of them, and add a linear layer to the decoder. MNIST samples generated with our approach are given in Figure 7. With a latent space of dimension 2, the MCNF generates examples that resemble real digits. Interpolating between points in the latent space gives hybrid intermediaries that meaningfully represent semantic change (for instance, going between a "4" and a "7" produces instances of "9").

(a)

(b)

(c)

Figure 7: (a) Real sample images from (Binarized) MNIST. (b) Random generated samples from a hyperbolic MCNF trained on Binarized MNIST with latent dimension 2. (c) Generated samples from the same trained MCNF, where the latent variables are taken from the projection onto $\mathbb{H}^2$ of a uniformly spaced grid on the tangent space $\mathcal{T}_0\mathbb{H}^2$. The generated samples are visualized in their corresponding positions on $\mathcal{T}_0\mathbb{H}^2$.