[Reviews · NeurIPS 2020]

Review 1

Summary and Contributions: This works generalizes Neural ODEs to be constrained on a manifold. Part of the motivation is being able to define invertible transformations that is directly applied to the manifold, instead of previous works that project a Euclidean normalizing flow to the manifold. The paper then shows how the resulting gradient computation includes an additional pushforward to map vectors of the manifold to vectors of (Euclidean) parameter space. The authors introduce a numerical technique where they map to Euclidean space, solve the ODE in Euclidean space, then map back onto the manifold. This reduces the number of exponential map evaluations compared to invoking it for every step of the solver. This was done for computational reasons. Finally, they show that the change of variables formula will include the extra pushforward within the trace, showing how log probability can be computed for manifold continuous normalizing flows. -- Update -- Thank you for your rebuttal.

Strengths: The approaches seem sound and are empirically validated. The approaches introduced in the paper are straightforward, easy to apply, and should be useful for many future works.

Weaknesses: Though authors repeatedly claim better tractability for higher dimensions, this is not tested experimentally. Only dimensions of up to 6 were experimented based on previous works. Furthermore, the log likelihood values in Table 1 are a bit worse than those reported in prior open source work [1]. Regarding the claims of generality, would it be incorrect to say works that project to/from the manifold using exponential maps / charts are equally general? Since it seems the main constraint on the set of applicable manifolds is whether the exponential map is tractable. [1] Bose et al. “Latent variable modelling with hyperbolic normalizing flows”.

Correctness: Prop 5.1 is missing a quantifier for epsilon. Is it for all or for some? If for some, I’d like to see a clarification on whether it’d be possible to derive the bounds on epsilon, and how/whether this is taken into account during the numerical integration procedure.

Clarity: The work is generally well written and I enjoyed reading it. The background section especially is clear and concise, which should help this paper reach a larger audience. I was confused by some parts of the motivations. The paper initially explains the downsides of projecting between the manifold and Euclidean space, and modeling transformations in Euclidean space. But the dynamic chart method seems to do precisely this. If I understand correctly, standard invertible transformations can also be applied locally, and stacked with multiple charts.

Relation to Prior Work: Yes.

Reproducibility: Yes

Additional Feedback:


Review 2

Summary and Contributions: The paper introduces Neural ODEs on manifolds. The approach uses also local geometry and can be applied to s wider range of geometric spaces compared to the previous approaches. The authors also extend the Instantaneous Change of Variable theorem to the arbitrary manifolds, allowing to perform density estimation. ===================================== Added after author response: The authors have addressed my concerns about the choice of epsilon, runtime and generality. The paper provides a valuable contribution to the field of Neural ODEs and normalizing flows. My score (7: accept) remains the same.

Strengths: The authors provide a solid theoretical analysis of their approach, including the correctness and convergence properties. They describe in detail how the approach can be used on different types of manifolds. I appreciate that the authors addressed the Related work in detail and built up a strong motivation for their work.

Weaknesses: The Dynamic Chart Forward Pass requires to solve the ODE locally within the epsilon proximity of the current point. Is there an estimate of how small the epsilon should be? Will the approach suffer from using larger epsilon? If epsilon needs to be small and fixed, Neural ODE becomes equivalent to the Euler solver. Does the method really benefit from using more complex solvers and the adjoint method in this case? Since MODE requires a series of small steps, I expect it to bequite slow. How does the runtime compare to other approaches, namely, CNF and PRNVP? In Dynamic chart method, how is the method influenced by the choice of the charts and their ordering in the eq. (3)?

Correctness: The claims and the methodology are correct.

Clarity: The paper is clearly written and provides both theoretical justification and the intuition behind the method.

Relation to Prior Work: This paper gives an extensive explanation about the differences to the previous works.

Reproducibility: Yes

Additional Feedback: One question that remains unclear to me is the choice of the charts for non-Riemannian manifolds. Is it performed using the trivialization technique by Lezcano-Casado et al? If so, the authors might want to include the overview of the technique in the paper. Although this paper addresses an interesting problem, I am wondering about the potential practical use-cases of the Neural ODEs for manifolds. The "Broader impact" section refers to the "applications in physics, robotics, and biology", but don't elaborate on it. Can you give an example?


Review 3

Summary and Contributions: The paper propose Neural Manifold Ordinary Differential Equations (NMODE) which generalize the Neural ODEs to Riemannian manifolds. The paper claims: 1) NMODEs generalise to arbitrary manifolds instead of making use of handcrafted layers and outperform handcrafted normalising flows on their specific domain. 2) NMODEs generalise NODEs to manifolds, deriving a manifold version of the adjoint method. == Post rebuttal == I have read the rebuttal, and my score remains unchanged. I think the paper is worth publication, but I had wished for a stronger rebuttal. The authors skipped answering my key concern: that we need to evaluate the determinant of the Jacobi field. The authors merely state that this is always possible, which, while correct, completely misses the point. The concern is that such an evaluation quickly becomes prohibitively expensive. The approach is really only feasible on simple manifolds where closed-form expressions are available or on sufficiently low-dimensional manifolds where the brute-force approach is computationally tractable. I still think the paper has merit, but I strongly encourage the authors to discuss this issue in the paper.

Strengths: It is valuable with a more general method for defining flows on manifolds. Both method and theoretical analysis is novel, albeit somewhat incremental.

Weaknesses: While the method is general, its implementation does not appear to be so. As far as we can tell, it may be non-trivial to extend beyond the studied spheres and hyperbolic spaces. This is concerning as part of the contribution is the generality of the method. To be a bit more specific (we encourage the authors to correct us if there are misunderstandings): (1) It seems (lines 265-266) that the determinant of the derivative of the exponential map (Jacobi field) is needed. In principle this can be evaluated on any manifold, but practically speaking this may be difficult beyond the most simple manifolds (as those studied in the paper). (2) The paper makes a distinction between local diffeomorphisms and the exponential map, which is quite interesting when considering general manifolds. However, as far as we can tell, in practice these local diffeomorphisms are always chosen to be the exponential map.

Correctness: The work appears to be correct.

Clarity: The paper is well written and reasonably easy to follow. Our only concern regarding the writing is that some issues appear somewhat "swept under the rug", so to say. For instance, relying on Jacobi fields is potentially very limiting, yet this is only something that is mentioned as a minor remark. Likewise, if the interpretation that exponential maps are used whenever local diffeomorphisms are required is correct, then this should be stated much more explicitly. It is understandable that the authors want to claim generality, but this should not be taken too far.

Relation to Prior Work: The paper does a nice job of positioning the work and the experiments are well-aligned with this discussion.

Reproducibility: Yes

Additional Feedback: Is it correct that Theorem 4.1 follows almost immediately from the corresponding result from [4] due to the ambient assumption? The comment in lines 190-191 regarding the Whitney embedding is not entirely convincing as the Whitney embedding Theorem is not isometric. That is, the method should be applicable, but it might imply a change of metric (and hence measure). The general claim may be correct, but a bit more detail would be helpful. EXPERIMENTS ============== The experimental section is what you'd expect, i.e. density estimation on the sphere S^2 and in hyperbolic space H^2 tested against 2 flow baselines and variational inference on the MNIST dataset with a fixed hyperbolic manifold as a latent space tested against Euclidean models (VAE + VAE with 2 flow variants) on the one hand and models handcrafted for hyperbolic space (hyperbolic VAE and VAE with hyperbolic normalizing flows among others) on the other. The latent dimensions tested were {2, 4, 6}. The method proposed mostly outperforms the baselines but not by a large margin and already by dim=6 this advantage ranges from negligible to non-existent, even against Euclidean methods, which we have also seen happen to one degree or another in our own work. I'd like to have seen a more fleshed out discussion of the results as well, for example their take on why the above happens. ** small corrections ** ——————————— Line 121: D_{x}f: T_{x}M \rightarrow T_{x}N —> D_{x}f: T_{x}M \rightarrow T_{f(x)}N Line 232: “[…] and the manifold” —> “[…] to the manifold”


Review 4

Summary and Contributions: The authors propose the Neural Manifold Ordinary Differential Equations to construct the manifold continuous normalizing flows. It only requires local geometry and thus can be generalized to arbitrary manifolds.

Strengths: The paper is based on solid knowledge about differential geometry. By using the local charts, the manifold ODE can be solved in the embedding space of the local charts, which further enables the construction of the manifold continuous normalizing flow. Introducing the charts of the manifold to manifold ODE is a good idea.

Weaknesses: For the manifold defined by high dimensional point cloud like images, computing the local charts of the manifold itself is hard and time consuming. The examples are simplified cases. For the experiments, it is unclear whether VAE can learn the local chart structure and the Riemannian metric.

Correctness: Theoretically, the ODE on manifold setting is classicial in differential geometry.

Clarity: The paper is well written, all the concepts are clear. But the way to construct the local charts is unclear.

Relation to Prior Work: The work describes the difference. Conventional methods focus on ODEs on Euclidean space, this one focuses on that on manifold with multiple charts.

Reproducibility: Yes

Additional Feedback: [POST REBUTTAL] I thank the authors for the detailed response. There are still some concerns: 1. The work focuses on manifold ODEs. In this case, the probabiity density change should reflect the change of volume element, which is determined by the Riemannian metric tensor. This is one of the key difference from Euclidean case. If this is discussed more explicitly, it will be more helpful. 2. In general, the manifold structure should be given by the data set itself, in this work it is given as a priori. Furthermore, the latent space is modeled as a hyperbolic disk for MNIST, and use the Poincare disk model. It is more direct to use a Euclidean disk as the latent space in this senario. 3. The normalized flow is a sequence of diffeomorphisms, the product of the Jacobians gives the ratio between the target and source densities. But this can be easily achived using optimal transport theory using convex optimization, which ensures the existence and uniqueness. Some comparisions will be helpful. 4. More importantly, the flow maps one simple distribution to the data distribution in the latent space, as shown in the MINIST example. But the support of MINIT latent distribution has multiple connected components (10 modes), the support of the source measure is simply connected, there is NO diffeomorphisms between them. Namely, the diffeomorphism will map part of the source measure to the gaps among the modes of MNIST, this will cause mode mixture or mode collapsing. This needs to be further discussed.

[Author Response · NeurIPS 2020]

We thank all reviewers for their time and constructive comments.

**Common Concerns.** We first address concerns that were brought up by multiple reviewers.

- **(R 1, 2) Tractability and Runtime**. Since we use the CNF/ Neural ODE solver, our runtime is comparable to
that of CNF, and thus we can use advances such as FFJORD [18] to scale with manifold dimension. Also note
the log-determinant formulas for tested manifolds scale linearly in dimension, giving us tractability. PRNVP
is faster as it uses the (non-continuous) RealNVP in the projected space. Also, our experiments show that
NMODE is more sample efficient than other methods (Appendix C.2, first paragraph), so for density estimation
we can use substantially fewer batches than other methods; this results in lower overall runtime.

- **(R 1, 2) Choice of** $\epsilon$. Appendix D.1 discusses controlling the dynamics with $\epsilon$. With a bound on the derivative
we enforce a Lipschitz constraint on the NMODE solutions, in which case $\epsilon$ can be chosen less than $r/L$,
where $L$ is the Lipschitz constant and $r$ is the local radius of injectivity. $\epsilon$ can be arbitrary for hyperbolic
space (since the radius of injectivity is infinite) and should be less than $\pi/L$ for the sphere. We find that most
reasonably small choices work in practice. The quantifier for Prop 5.1 should be "for some"; this will be fixed.

- **(R 2, 3, 4) Applications and Concerns about Results** First note that aside from the general use cases
presented in the paper (to data that does not require a manifold constraint), our approach gives a way to learn
manifold-constrained flows for problems that require this constraint. In physics, we allow learning tractable
densities on unitary Lie groups for Lattice Quantum Field Theory [37], in biology we need these densities
for protein-structure prediction [21], and in robotics we need these for path navigation/motion estimation
(e.g. modeling motion of a robot arm as a density in $\mathbb{T}^6$) [13]. Regarding our results, the lower margins with
respect to Euclidean baselines as we increase dimension are due to the fact that it is possible to compensate for
geometric distortion in higher Euclidean dimensions (Nickel et al., 2017). Note that for small dimensions (e.g.
2) our results significantly exceed the baselines and this is important for application via faster inference (more
efficient modeling of data in low dimensions). We also note that although the example densities are simplified
cases, we outperform all state-of-the-art baselines, highlighting existing flaws that we resolve.

- **(R 2, 4) Choice and Construction of Local Charts** The manifold is determined a priori in many applications,
as in all applications above. Thus the local charts are determined a priori when the requisite manifold is
selected and the construction follows from the manifold definition. Regarding "...choice of the charts for
non-Riemannian manifolds...", note that the manifolds considered in our paper are smooth, hence admit a
Riemannian metric, and are thus Riemannian. Therefore all manifolds we consider have well-defined charts.

**Reviewer 1** Thank you for your kind words on our approach and the potential future impact of our paper.

- **"...the log likelihood...worse than those reported in...[2]."** We used [2]'s code (we had access before they
open sourced it), fixed bugs in their implementation (that the authors confirmed to be actual bugs), and also fix
evaluation (we run with a validation set, unlike [2], which gives misleadingly high numbers).

- **"Regarding the claims of generality..."** and **"I was confused by some parts of the motivations."** Yes, both
(exp map and chart) are equally general. Projecting from the ambient space is not as principled (projecting $\mathbb{R}^3$
to $\mathbb{S}^2$ is not well-defined) whereas our exponential map helps learn over the space directly. We claim to be the
first general principled method; we will augment the paper to make this more clear.

**Reviewer 2** We are glad you appreciated the theoretical aspects and motivation of our paper.

- **"Does the method really benefit from using more complex solvers and the adjoint method in this case?"**
Note that other methods do not provide the ability to incorporate more complex solvers from the classical
literature, yet we do. Certainly, switching to one of these solvers will not always help (as in the case you
mentioned). However, $\epsilon$ can frequently be fairly large; in these cases, a more complex solver would help.

**Reviewer 3** Thank you for recognizing our theoretical and algorithmic merits.

- **"determinant of the derivative of the exponential map is needed...this may be difficult beyond the most**
**simple manifold..."** This is possible given a parameterization of the manifold via local charts.
- **"...exponential maps are used...this should be stated much more explicitly..."** This will be done.
- **"Thm 4.1 follows from [4] due to the ambient assumption?"** Yes, the proof is analogous to that of [4].
- **"...the Whitney embedding Theorem is not isometric..."** NMODE itself is not affected by the change of
metric. For MCNF, note that for relevant manifolds (e.g. those for applications given above: hyperboloid,
sphere, unitary Lie groups, etc.) the data has an explicit embedding.

**Reviewer 4** Thank you for your comments and appreciation of our usage of charts.

- **"...whether VAE can learn the local chart structure and the Riemannian metric..."** The VAE does not
learn the manifold structure; the hyperbolic manifold is chosen a priori.

[Meta-Review · NeurIPS 2020]

Good paper that extends continuous-time flows from Euclidean spaces to manifolds. The paper is an important contribution that moves the field of normalizing flows forward. Well done. One reviewer raised the concern that the determinant of the Jacobi field could be prohibitively expensive to evaluate. I would encourage the authors to clarify this point in the camera-ready, and discuss whether/when it would be the case. If indeed this is the case, this is an important issue to acknowledge in the paper, that could be useful in pointing researchers to directions for future work. I would also encourage the authors to acknowledge in the camera-ready the parallel work by Falorsi & Forré (https://arxiv.org/abs/2006.06663), and mention how it's similar and how it differs.